# Social Distancing Policies in the Coronavirus Battle: A Comparison of Denmark and Sweden

**DOI:** 10.3390/ijerph182010990

**Published:** 2021-10-19

**Authors:** Ida Seing, Nina Thórný Stefánsdóttir, Jeanette Wassar Kirk, Ove Andersen, Tine Tjørnhøj-Thomsen, Thomas Kallemose, Evert Vedung, Karsten Vrangbæk, Per Nilsen

**Affiliations:** 1Department of Behavioral Science and Learning, Linköping University, SE 581 83 Linköping, Sweden; 2Department of Clinical Research, Copenhagen University Hospital—Amager and Hvidovre, DK-2650 Hvidovre, Denmark; nina.thorny.stefansdottir@regionh.dk (N.T.S.); jeanette.wassar.kirk@regionh.dk (J.W.K.); ove.andersen@regionh.dk (O.A.); thomas.kallemose@regionh.dk (T.K.); 3Department of Public Health, Nursing, Aarhus University, DK-8000 Aarhus, Denmark; 4Department of Clinical Medicine, University of Copenhagen, DK-2200 Copenhagen, Denmark; 5National Institute of Public Health, University of Southern Denmark, DK-1455 Copenhagen, Denmark; titt@sdu.dk; 6Institute for Housing and Urban Research (IBF), Uppsala University, SE 751 20 Uppsala, Sweden; evert.vedung@ibf.uu.se; 7Department of Government, Uppsala University, SE 751 20 Uppsala, Sweden; 8Department of Public Health, University of Copenhagen, DK-1353 Copenhagen, Denmark; kv@ifs.ku.dk; 9Department of Political Science, University of Copenhagen, DK-1353 Copenhagen, Denmark; 10Department of Health, Medicine and Caring Sciences, Linköping University, SE 581 83 Linköping, Sweden; per.nilsen@liu.se

**Keywords:** policy measures, social distancing, laws, executive orders, recommendations, comparative research, document analysis

## Abstract

Social distancing measures have been a key component in government strategies to mitigate COVID-19 globally. Based on official documents, this study aimed to identify, compare and analyse public social distancing policy measures adopted in Denmark and Sweden regarding the coronavirus from 1 March 2020 until 1 October 2020. A key difference was the greater emphasis on laws and executive orders (sticks) in Denmark, which allowed the country to adopt many stricter policy measures than Sweden, which relied mostly on general guidelines and recommendations (sermons). The main policy adopters in Denmark were the government and the Danish Parliament, whereas the Public Health Agency issued most policies in Sweden, reflecting a difference in political governance and administrative structure in the two countries. During the study period, Sweden had noticeably higher rates of COVID-19 deaths and hospitalizations per 100,000 population than Denmark, yet it is difficult to determine the impact or relative effectiveness of sermons and sticks, particularly with regard to broader and longer-term health, economic and societal effects.

## 1. Introduction

The global spread of severe acute respiratory syndrome coronavirus 2 (SARS-CoV-2), henceforth the coronavirus, is the defining global health crisis of our time. On 11 March 2020, the World Health Organization (WHO) declared the outbreak of the coronavirus as a pandemic and a public health emergency of international concern [1]. Globally, social distancing became the main strategy to limit the spread of the coronavirus, based on the premise that the virus’ spread would slow down if citizens stayed home from work or school, avoided large gatherings and refrained from touching one another [2]. However, although social distancing measures have been a key component in government strategies to mitigate pandemic influenza globally, there is relatively limited evidence of the effectiveness of many social distancing measures; many initiatives are described in the research literature as being moderately effective [3,4,5].

Many different social distancing strategies have been used internationally in the battle against the coronavirus, from wide-reaching public policy measures, such as nationwide compulsory lockdowns of large sectors of the society, to more limited voluntary measures, such as recommendations to keep physical distance in shops and restaurants. The two neighbouring countries, Denmark and Sweden, differed in their national response to the coronavirus pandemic. Denmark was described in the media as an early mover and was seen as something of a test case in its swift response, with mandatory social distancing measures [6]. A number of restrictions were in place early in the pandemic, including closure of the national border, bars and restaurants and limiting gatherings to 10 people [7].

In contrast, Sweden’s strategy was characterized by voluntary measures, with an emphasis on people’s willingness to assume responsibility for social distancing [8]. Although Sweden never targeted herd immunity as a declared strategy, the country was critiqued for what many perceived as a relaxed strategy that allowed the spread of the virus throughout the population [9]. International criticism of Sweden was exacerbated by the fact that the country had far higher COVID-19 incidence, hospitalization and mortality rates than Denmark [10].

Denmark and Sweden share many economic, political and cultural characteristics [11], which begs the question of why the countries chose such different social distancing strategies, as conveyed in the media. The disparate governmental strategies adopted by the two countries offer a unique opportunity to study public policy measures embraced to achieve social distancing. A comparative analysis of Denmark’s more common strategy and Sweden’s more unusual strategy can contribute to improved understanding of the factors that influence government adoption of policies, which is particularly relevant due to the limited evidence base for many social distancing measures. We have not been able to find any comparative research on the social distancing policy measures adopted by Denmark and Sweden to tackle the coronavirus. Overall, this study contributes with important knowledge about different national responses to a societal crisis, and how and why the strategies may differ between countries. The aim of this study is to identify, compare and analyse social distancing policy measures regarding the coronavirus that were adopted in Denmark and Sweden in 2020.

## 2. Materials and Methods

### 2.1. Study Design and Definitions

The study is based on a document analysis [12] of officially published documents describing the social distancing public policy measures adopted in the two countries during the initial pandemic period, over the course of seven months from 1 March 2020 until 1 October 2020.

The documents are official versions of issued social distancing policy measures, such as laws, executive orders and general guidelines. In addition, the documents consist of press releases and news about the issued policy measures from authorities, governments and parliaments in the countries. The documents were retrieved from the websites of relevant authorities, governments and parliaments and official websites (owned by governmental authorities) that publish current national legislations in the two countries.

Table 1 provides information about document type and which organizations and websites the documents were predominantly retrieved from in Denmark and Sweden.

The focus of the study is on the formal decisions concerning public policy measures. We have not included various forms of practice-oriented guidelines developed by public authorities. The WHO advocates the use of the term “physical” distancing rather than “social” distancing because it more accurately reflects the practices involved, given that digital technology has enabled people to be socially connected without being physically close to each other [13]. In this article, we use the term social distancing because it is still widely used.

The study is based on a comparative approach, usually understood in social research as the analysis of similarities and contrasts among different macro-level units (countries, in this study), at one or more points in time [14]. Comparative research aims to reach more general conclusions beyond single cases (that is, Denmark and Sweden) and explain the differences and similarities against the backdrop of the contextual conditions of the cases.

Public policies are understood as formal decisions adopted by state organizations to achieve particular goals [15]; that is, compliance with social distancing in this study. Policy measures are the actions carried out to implement a public policy [16]. Policy measures can be distinguished into three general categories of sticks, sermons and carrots, in accordance with a taxonomy by Vedung [17,18]. Sticks (also referred to as regulations in Vedung’s taxonomy) are policy measures that mandate people to act in accordance with what is ordered. Sticks are mandatory for the target population, that is, they must be complied with by citizens and institutions. Sermons (also referred to as information) attempt to influence people through the transfer of knowledge, communication of reasoned argument and persuasion. Sermons are non-binding, voluntary rules and recommendations directed at citizens and organizations. Carrots (also referred to as economic means) are economic measures, such as subsidies and grants, involving provision or removal of material resources. The addressees of the two latter measures are not obligated to take the actions involved [17,18].

### 2.2. Study Procedure

Table 2 provides an overview of the data that were obtained. The aspects examined were selected because they are usually described as determinants of implementation activities and outcomes, that is, characteristics of the strategy or other means of facilitating implementation (general types of policy measures), characteristics of the intervention (categories of social distancing policy measures), characteristics of the decision-makers (policy adopters, time of adoption and implementers) and characteristics of the end users (target groups) [19,20].

The data collection and analysis process consisted of several steps. Initially, central actors (authorities, governments and parliaments) involved in adopting social distance policies were identified. Thereafter, the policy measures (including the documents) were identified, retrieved and mapped onto a table/matrix based on aspects described in Table 2. The documents (table/matrix) in each country were continuously updated and revised as new measures were identified. During this process, the documents were discussed among representatives for the two groups from Denmark and Sweden in several Zoom meetings and e-mails to ascertain consistency in terminology. The two documents were compared, analysed and discussed by all researchers involved in the study in Zoom meetings.

### 2.3. Study Context

Denmark and Sweden are neighbouring Scandinavian countries with many similarities, although they differ with regard to some features of potential relevance for understanding their respective responses to the pandemic. The histories of the two countries are intertwined because they were united under the same rule for more than a century. Both are modern, stable, non-corrupt and prosperous welfare states with large public sectors. They share a similar socio-cultural heritage, including similar languages. Denmark and Sweden are both constitutional monarchies with limited power for the ruler; power is exercised through parliaments, governments and ministers.

Like most western democracies, Denmark has a system of ministerial governance, whereby the public administration is led by ministries that have full responsibility for public authority activities and the mandate to make many decisions [21]. The public administration is cohesive, and the authorities are formally part of the ministerial hierarchy. However, the role of the authorities is also to provide input to decision-making based on independent professional expertise and assessments of evidence [22,23].

A distinguishing feature in the Swedish Administrative Model is that of independent public authorities, where individual minsters cannot issue orders to the public authorities to act on a decision; only the government as a whole can issue such orders and only in general form [24,25]. Compared with Denmark and most western democracies (except Finland, which has a partially similar system), the public authorities in Sweden are organizationally separate from the central governmental ministries, with considerable legal autonomy. Formally, the authorities are subordinate to the government as a whole, but they have an independent position and are led by their own Director General [26]. Furthermore, individual government ministers in their comparatively small ministries cannot issue orders or advice to individual authorities; the power to do so rests with the government as a whole (government in corpore). Finally, not even the government as a whole can issue orders or advice to authorities concerning their application of general rules to individual cases; in other words, by constitutional fiat it is not allowed to interfere in cases of individual decision-making by authorities. Together, these three features constitute the gist of the independence of Swedish public authorities [27,28,29].

Both countries have publicly funded, decentralized and universal health and welfare systems, and every citizen has equal access to services. The system in both countries has three levels: the state, regions (5 in Denmark, 21 in Sweden) and municipalities (98 in Denmark and 290 in Sweden). The state in both countries establishes principles and guidelines for health and welfare and sets the political agenda by means of legislation. The regions are responsible for provision of health care (primary health care and hospital services) and have considerable freedom in planning for the delivery of care. Municipalities are responsible for elderly care, among other services.

## 3. Results

The results are presented with regard to the six aspects described in Table 1. The results are presented for Denmark and Sweden separately before they are compared.

### 3.1. General Types of Policy Measures

#### 3.1.1. Denmark

Three types of policy measures were identified to achieve social distancing: (1) laws (love); (2) executive orders (bekendtgørelser); and (3) recommendations and travel advice (anbefalinger, rejsevejledninger). Although a few laws were identified, most of the measures belonged to the two other categories; executive orders were the most common. The laws and executive orders were classified as sticks, whereas recommendations and advice were categorized as sermons.

Laws and executive orders (that is, sticks) consisted of mandatory elements with requirements for people or organizations to behave or act in a certain way to comply with the social distancing policy measures. The executive orders were associated with the laws.

Revision of the so-called Infectious Diseases Act was the most central law introduced as part of the COVID-19 crisis management. This act granted the Minister of Health a number of powers to ensure containment of dangerous diseases (such as COVID-19). For instance, this act included the possibility to lock down large parts of society and prohibit participation in and holding of larger events and assemblies. Executive orders were issued to ban visitors in long-term care facilities, such as nursing homes and hospitals as well as care facilities in the area of social affairs. In addition, the Infectious Diseases Act granted the Minister of Health the power to order persons who were suspected of being infected to be isolated, as well as the possibility of banning access to or imposing restrictions on means of transport.

Recommendations and travel advice were a voluntary measure (sermons) which provided guidance to organizations and/or individuals on how to act or behave to achieve social distancing. The language of these policy measures was instructive and normative; for example, posters offered advice to the Danish population to “self-isolate if you have symptoms, have tested positive or are a close contact of someone who is infected with novel coronavirus”, “do not shake hands, hug or kiss as a greeting—limit physical contact” and “keep your distance and ask others to be considerate”.

The three types of policy measures were distributed through several channels and in many different formats. For example, advice was communicated through press conferences, nationwide and local media platforms and various formats such as posters, pamphlets, digital banners, print ads, radio and TV spots, content for social media and e-mails in E-boks, an online digital mailbox for Danish citizens [30].

#### 3.1.2. Sweden

Three types of policy measures were identified to achieve social distancing: (1) laws (lagar), (2) executive orders (förordningar) and (3) general guidelines and recommendations (allmänna råd, rekommendation and reseavrådan). General guidelines and recommendations were the most common, followed by executive orders. Few adopted laws were identified. Laws and executive orders were classified as sticks, whereas general guidelines and recommendations were categorized as sermons.

Laws and executive orders consisted of mandatory elements with requirements for people or organizations to behave or act in a certain way to comply with the goals of social distancing. For example, if restaurants, bars and cafes did not follow the law on temporary infection control measures, municipalities had the authority to temporarily close their business. If organizers violated the ban on public gatherings or public events with more than 500 or, at times, 50 participants, the police had the right to cancel or dissolve activities that were held in violation of the regulation.

General guidelines and recommendations provided a form of detailed guidance to individuals and organizations on how to act or behave to achieve social distancing. These types of measures were not legally binding, because there were no penalties or negative sanctions if an individual or organization did not comply. Many of the measures were linked to laws and executive orders by means of providing clarifications on how these should be complied with. General guidelines often used terms such as “should” (“bör”) and “are encouraged to” (“uppmanas”).

The three types of policy measures were distributed through various communication channels, such as posters, information sheets, public agency home pages, television (press conferences with, for example, the Public Health Agency or representatives from the government) and social media, aimed at convincing organizations and the general population to comply with the measures.

#### 3.1.3. Comparison

Three types of policy measures were identified to achieve social distancing in both Denmark and Sweden. However, Denmark used a larger number of sticks (laws and executive orders) than Sweden, where general guidelines and recommendations were much more common. In both countries, policy measures were altered and updated several times as a form of ongoing formative evaluation in response to information about the spread of the coronavirus, for example, the number of people infected, the number of hospitalizations and mortality rates due to COVID-19. General guidelines and recommendations (sermons) in Sweden were transformed into executive orders (sticks), but this was not the case in Denmark.

### 3.2. Categories of Social Distancing Policy Measures

#### 3.2.1. Denmark

The categories of social distancing policy measures were largely consistent with the taxonomy by Rashid et al. [3]. When the lockdown in Denmark was initiated, schools and day care facilities were closed (school closure). Instructions for providing emergency teaching were given. Certain examinations were cancelled or re-organized as online or video examinations. Various workplace restrictions were put into effect (workplace closure and home working). Recommendations for home working for non-health care workplaces, such as increased use of tele-commuting and remote meeting options, were issued. In relation to the isolation of cases and quarantine of contacts, isolation facilities for citizens infected with coronavirus were established. Further, an executive order was issued with directions on isolation in case of contamination and assumed contamination. It was stated that anyone who had COVID-19 or who was assumed to be infected could be ordered to isolate themselves.

Measures were also introduced concerning mobility restrictions and cancellation of mass events. These included cancellation of and a ban on mass gatherings and closure of areas where many people gather. In relation to mobility restrictions, only travellers with a legitimate purpose could cross the Danish borders, and the Ministry of Foreign Affairs repeatedly updated their travel advice with cautions against unnecessary travel abroad. Furthermore, the landing of cruise ships was prohibited and a maximum occupancy in long-distance buses was announced. In addition, recommendations for hosting demonstrations and for hosting parties were given. Parents were advised on the arrangement of play dates, including advice to avoid play dates with and among too many children and a call not to host children’s birthday parties and sleepovers.

Policy measures not part of Rashid et al.’s taxonomy [3] included lockdown of various forms of public places and areas, such as closure of all sports facilities, fitness centres and solariums. In addition, hairdressers and other companies with close customer contact closed unless they performed health care services. All nightclubs, restaurants, cafes and so forth closed to guests (except for takeaway). All major shopping malls closed. There was also a temporary ban on visitors to long-term care facilities and hospitals.

#### 3.2.2. Sweden

All categories of Rashid et al.’s taxonomy [3] were identified, except workplace closure, which did not occur in Sweden. In relation to school closure, measures were taken regarding distance education for students at, for example, universities and high schools. Regarding home working, there were general guidelines and recommendations urging employers in Sweden to take responsibility to ensure that their employees, if possible, kept their distance from each other, worked from home, avoided unnecessary travel at work and could adjust their working hours to avoid travelling during rush hour. With regard to the categories of isolation of cases, self-isolation of cases and quarantine of contacts, policy measures, communicated as general guidelines, urged the population who were infected or had symptoms of COVID-19 to stay home and avoid social contacts.

With regard to mobility restrictions, policy measures were in place to restrict people’s movement within Sweden and abroad. General guidelines urged people >70 years of age to avoid using public transport, and this general advice was applied to the Swedish population at large during rush hours. Examples of policy measures concerning cancellation of mass events included bans on mass gatherings of more than 500 people (later adjusted to 50 people) and general guidelines urging the population to avoid participating in major social events, such as parties, funerals, baptisms and weddings.

Further policy measures that could not be categorized into Rashid et al.’s taxonomy [3] included various ways to ensure physical distance between people in public areas, for example, a recommendation for people to keep distance from each other indoors and outdoors in places where people gather, such as shops, shopping centres, museums, libraries, service offices and waiting rooms. There were also measures aimed directly at businesses, such as restaurants and shops, emphasizing their responsibility to ensure distance between guests and visitors. Additionally, there were measures to prohibit and limit visits to public institutions, such as elderly care homes and hospitals. Early in the outbreak of COVID-19, there were recommendations that people over the age of 70 years must limit their social contacts until further notice.

#### 3.2.3. Comparison

One difference between the countries was that the policy measures in Denmark consisted of restrictions, that involved various forms of closure of the society, to a greater extent than in Sweden. Thus, in Denmark, the policy measures consisted of lockdown and more micro-level governance. In both countries, most types of social distancing policy measures were consistent with Rashid et al.’s taxonomy [3]. However, in both countries, additional policy measures were identified that could not be categorized within the taxonomy. This applied, in particular, to measures concerning closure (in Denmark) and/or physical distance (in both Denmark and Sweden) of/at various public places, such as restaurants, shopping malls and sports centres.

### 3.3. Policy Adopters

#### 3.3.1. Denmark

The main formal policy adopters were the Danish Government and the Parliament, which hold the overall responsibility for adopting laws and executive orders. Several ministries were involved, including the Ministry of Health, the Ministry of Social Affairs and the Interior, the Ministry of Foreign Affairs and the Ministry of Justice. Advice and recommendations were primarily adopted by the Danish Health Authority (Sundhedsstyrelsen). The Danish Health Authority has a national responsibility for health issues and works to ensure good public health and uniform health care services across Denmark, including effective health emergency management [31]. The authority advises the Ministry of Health and other governmental, regional and municipal authorities on health and elderly care and provides recommendations, guidelines and action plans based on independent assessment of the available evidence [31].

#### 3.3.2. Sweden

The Public Health Agency (Folkhälsomyndigheten) was responsible for issuing general guidelines and recommendations, which constituted the majority of the policy measures in Sweden. The authority is an expert state authority under the Government’s Ministry of Social Affairs, with national responsibility for public health issues. The authority aims to protect the population against communicable diseases and other health threats [32] and has a responsibility to develop general guidelines and recommendations. General guidelines are based on preparatory work for laws and the accumulated expert and legal knowledge in the field, and are developed by organizations and state authorities affected by the rules [33].

Laws and executive orders were adopted by the Swedish Government and the Parliament. The ministries involved included, for example, the Ministry of Social Affairs, the Ministry of Education, the Ministry of Justice, and, on recommendations concerning (dissuasion) to travel abroad, the Ministry of Foreign Affairs.

#### 3.3.3. Comparison

Reflecting Denmark’s emphasis on laws and executive orders, the key policy adopter was the government and its ministries as well as the Danish Parliament. The Government and Parliament in Sweden were much less publicly active in adopting policies and measures. Instead, the Public Health Agency had a central role in issuing general guidelines and recommendations.

### 3.4. Time of Adoption

#### 3.4.1. Denmark

Most of the policy measures in Denmark regarding social distancing were adopted in March 2020 when lockdown of larger parts of society began. On 6 March 2020, the Danish Prime Minister held her first press conference and called for the postponement or cancellation of all events with more than 1000 participants. This was followed by more press conferences. At a press conference on 11 March 2020, a partial lockdown (see examples of measures in the previous section) was announced, starting two days later; schools, day care centres and institutions were closed and an assembly ban for more than 100 people was passed. The government followed this up with further restrictions, such as the closure of restaurants, shopping centres and hotels, as well as closure of the borders. On 12 March 2020, the Danish Parliament adopted an amendment of the most central law related to social distancing, the Infectious Diseases Act, which authorized the Minister for Health to take measures to combat the COVID-19 pandemic.

By mid-April 2020, Danish society gradually re-opened with the physical opening of day care centres, primary schools and later universities, as well as other facilities, such as workplaces, cultural and leisure activities and restaurants. The re-opening was the result of a series of decisions announced at press meetings throughout the spring and early summer.

During the study period, many of the policy measures (executive orders) were altered and updated several times. Changes were typically aligned with the rise and fall in the infection and mortality rates as well as adjusted to the phased re-opening of society. For example, municipality policy measures were put into place to prevent further spread of the coronavirus when an increase in the number of cases was detected in certain areas of the country. In addition, the ban on public gatherings was increased from 10 to 50 people, but later reverted back to 10.

#### 3.4.2. Sweden

Most of the policy measures were adopted between March and the beginning of April 2020. Most were general guidelines and recommendations formally issued by the Public Health Agency aimed at the general public and organizations. On 1 April 2020, new instructions and general guidelines were issued by Public Health Agency [34]. Some laws and executive orders also emerged in March–April 2020. For example, on 11 March 2020, an executive order was adopted by the Swedish Government concerning a ban on public gatherings or public events with more than 500 participants.

Numerous policy measures were adjusted in response to how the coronavirus spread developed, and revised, extended or updated several times. For example, the prohibition of visits to nursing homes was changed from a recommendation (sermon) to an executive order (stick). The executive order on prohibition of public gatherings or public events with more than 50 participants was changed regarding the number of people allowed to participate (from 500 participants to 50 participants). The temporary ban on unnecessary travel to Sweden from countries other than the EU was extended several times.

#### 3.4.3. Comparison

In both countries, March 2020 was the most intensive month with regard to adopting policy measures. A marked difference, however, was that laws and executive orders (sticks) were dominant early in the pandemic in Denmark, whereas voluntary guidelines and recommendations (sermons) were most common in Sweden. Denmark was earlier than Sweden in adopting more strict policy measures to strengthen social distancing, including lockdowns.

### 3.5. Policy Implementers

#### 3.5.1. Denmark

The organizations responsible for implementing the adopted policy measures were generally public authorities directly organized under the ministries, for example, the Danish Health Authority and the Danish Patient Safety Authority as well as the State Serum Institute, but organizations such as municipalities (as well as municipal boards), including day care facilities, educational institutions, residential institutions and unemployment offices, and regions (as well as regional councils) including hospitals, were also involved. Furthermore, implementers included private actors, such as transport companies, organizers of events and administrators of airports and ports as well as various non-governmental organizations.

The police were the main organization responsible for monitoring and controlling compliance with the policy measures, but the Danish Defence Forces, municipalities and the Danish Patient Safety Authority also had roles in these efforts. These actors had the authority to impose sanctions if the laws and executive orders were not complied with.

The Danish Health Authority and the Danish Patient Safety Authority were the most central organizations providing support to facilitate compliance with the measures. The Danish Health Authority generated information materials, such as posters, pamphlets and TV and radio items.

#### 3.5.2. Sweden

Organizations charged with implementing the adopted policy measures included both public and private actors, such as municipalities, public authorities, schools, elderly care, hospitals, public transport, employers, businesses and organizers of sporting events, concerts, cinemas and so forth.

Organizations with responsibility for monitoring and controlling compliance with the policy measures included the police, municipalities and infection control physicians, that is, actors with the authority to impose sanctions if the laws and executive orders were not complied with. For example, if restaurants, bars and cafes did not follow the law on temporary infection control measures, municipalities had the authority to temporarily close their business. If organizers violated the ban on public gatherings or public events with more than 500/50 participants, the police had the right to cancel or dissolve public gatherings and public events that were held in violation of the regulations.

The Public Health Agency, the main organization with the responsibility for providing support to facilitate compliance, provided miscellaneous guidelines, checklists and information materials to relevant organizations, such as schools, health care facilities and restaurants and businesses. Another public authority with a supporting function was the Swedish Civil Contingencies Agency. The National Board of Health and Welfare played an important role in terms of providing guidelines, checklists and information materials to organizations within health care and social services (including elderly care) in Sweden. These three state authorities held joint press conferences on Swedish national television concerning the current situation of the pandemic in Sweden.

#### 3.5.3. Comparison

Policy implementers in Denmark and Sweden had three roles: responsibility for implementing policy measures, monitoring and controlling compliance with the measures and providing support to facilitate compliance with the measures. In both countries, the implementers were actors from a broad range of society, within both public and private organizations. Many more policy measures were mandatory in Denmark compared with Sweden, which meant that implementers in Denmark, to a larger extent than in Sweden, consisted of organizations with more of a monitoring and controlling function, for example, the police and the Danish Defence Forces.

### 3.6. Policy Target Groups

#### 3.6.1. Denmark

The social distancing policy measures were targeted at the population at large. Measures were also directed at specific professions, such as health care professionals employed in hospitals and nursing homes and professionals within social affairs, education and day care, as well as the citizens utilizing these services, for example, patients, residents in nursing homes, university students and unemployed citizens. Furthermore, measures were targeted at people with increased risk of severe illness from COVID-19 and at citizens infected with coronavirus or with symptoms, or who had been in contact with a person who has been confirmed infected with coronavirus.

#### 3.6.2. Sweden

In Sweden, the social distancing policy measures were targeted at the population at large. Measures were also directed at specific sub-groups of the population, such as people >70 years of age, risk groups, residents in nursing homes, travellers, employees in public and private organizations and students in high schools, universities and colleges.

#### 3.6.3. Comparison

The policy targets were broadly similar in Denmark and Sweden, encompassing both the general population and specific sub-groups of the population.

## 4. Discussion

The comparative analysis identified both differences and similarities in governmental responses to COVID-19 in Denmark and Sweden. A key difference is the larger emphasis on sticks in Denmark and the stronger reliance on sermons in Sweden. This meant that Denmark had greater involvement of policy implementers with a monitoring and controlling function to ascertain that the measures were complied with. The use of laws and executive orders allowed Denmark to adopt stricter policy measures than Sweden. In contrast, Sweden predominantly relied on voluntary policy measures that encouraged people and organizations to behave in accordance with general guidelines and recommendations issued by the Public Health Agency.

The difference in emphasis on sticks versus sermons between Denmark and Sweden largely reflects a difference in political governance and administrative structure between the two countries. Recent studies on Sweden’s response to the pandemic attribute the use of recommendations and the Public Health Agency’s central role to the Swedish Administrative Model [35,36]. Public authorities in Sweden, including the Public Health Agency, are relatively independent of the government (in comparison with most other western countries), whereas public authorities in Denmark are led by a minister who is formally directly responsible for the authority’s activities and has the authority to make many decisions. Ministerial rule is prohibited in Sweden, whereas the minister in Denmark has the power to intervene and steer the everyday work of the authority, although ministers also have to respect the role of the authority as an independent source of expert advice [23].

The differences between the countries were most obvious in the early period of the coronavirus pandemic, when the Danish Government issued lockdowns of parts of society while Sweden remained open. Lockdowns in Denmark were made possible due to a revision of the so-called Infectious Diseases Act, in which the Minister of Health was granted a number of powers to ensure containment of dangerous diseases (such as COVID-19).

During the study period (from 1 March 2020 until 1 October 2020), Sweden had noticeably higher rates of COVID-19 deaths and hospitalizations per 100,000 population.

It is evident from Figure 1 that Sweden experienced considerably higher COVID-19-related death rates (per 100,000 population) than Denmark during the study period. Still, it is noteworthy that the difference was small towards October 2020. The overall picture is similar with regard to COVID-19-related hospitalizations (per 100,000 population), as shown in Figure 2. Sweden had higher per-capita death and hospitalization rates due to COVID-19 than most other European countries [10]. When the second wave of coronavirus hit in the autumn/winter 2020–2021 (i.e., after the study period), Sweden’s strategy changed in response to high and increasing rates of COVID-19 cases. A Pandemic Law was developed, and many more sticks were adopted by the Swedish Government, thus making Sweden’s strategy more similar to Denmark’s. It was debated in Sweden whether this change was due to the recognition that sermons were not sufficiently effective to reduce the spread of the virus [9]. There is no question that Denmark fared better in the short run (during the study period), as measured by numbers of COVID-19-related deaths and hospitalizations per capita. However, it is difficult to determine the impact or relative effectiveness of sermons and sticks, particularly with regard to broader and longer-term health, economic and societal effects [37].

Sweden’s deviating response and high rates of COVID-19-related per-capita deaths and hospitalizations generated considerable debate in Sweden as to whether a lockdown would be more effective than the chosen strategy to reduce the spread of the virus. Public health and medical experts as well as miscellaneous researchers seemed to be divided between those who advocated stricter restrictions (for example, closure of schools and public areas) and those who supported the chosen strategy. At the Public Health Agency, experts expressed concerns as to whether a societal lockdown, as implemented in Denmark, would result in increased spread of the virus when the countries re-opened again. Experts at the Public Health Agency were motivated in their recommendations in terms of the need for longer-term, sustainable restrictions because they expected COVID-19 to be around for a long time [9].

Overall, the Swedish Government had a limited role as a formal policy adopter, since most policy measures to achieve social distancing were based on general guidelines and recommendations issued by the Public Health Agency. Thus, the Swedish strategy during the study period seemed to rely more on expertise than on politics, that is, primarily on civil servants at the Public Health Agency, who issued general guidelines and recommendations as well as communicated with and provided advice to the politicians on suitable policy measures. In contrast, the Danish Government had a central role as policy adopter in Denmark and, in some cases, adopted stricter measures than recommended by experts at the Danish Health Authority by referring to information obtained from international experts and international agencies [38].

As a consequence of the reliance on the Public Health Agency, the Swedish Government was criticized for its lack of leadership in the coronavirus pandemic [9]. The pandemic raised questions about whether the Swedish Administrative Model was really suited for major societal crises [39,40,41]. In the media and academic debate, a common argument was that effective crisis management was hindered by the extensive autonomy afforded the national public health authorities (including the Public Health Agency) and the country’s 21 self-governed regions, which are responsible for providing health care in Sweden. It was argued that this model works well under normal circumstances, but it could be ineffective in crisis situations when rapid decisions need to be taken and implemented [39].

Despite the different strategies in response to the spread of coronavirus, our study findings also point to many similarities in the responses of the two countries, for example, regarding categories of policy measures for social distancing, policy adopters in the crisis management and implementers and target groups of the measures. Although the Danish Government and Parliament were behind a large number of policy measures, our findings suggest that public health authorities in both countries were important policy adopters in terms of issuing recommendations and advice to the population and organizations. In both countries, the national public health authorities performed a key role in adopting voluntary policy measures (sermons) and in communicating with the public by means of representatives giving press conferences and appearing on national television. It is illustrative of the differences that the Danish press conferences featured experts from the relevant authorities together with politicians, whereas the Swedish press conferences almost exclusively relied on the Public Health Agency experts (together with other relevant authorities).

There was a public debate in Sweden whether compliance with the sermons-oriented strategy would be facilitated by the interpersonal and institutional trust of citizens [9]. Research has documented that the level of trust in a society can have considerable impact on the conditions for crisis management in a health crisis, implying that the same types of policies (for example, social distancing policy measures) can have different impacts depending on the levels of trust [42,43]. The role of public trust during pandemics has been highlighted as a crucial factor in achieving compliance with adopted measures among populations [43,44]. However, the levels of institutional and interpersonal trust are broadly similar in Denmark and Sweden; all the Nordic countries have high levels of trust compared with most other countries in the world [45,46]. Thus, the high level of trust may be a contributing factor to explain why the Nordic countries have performed relatively well internationally in battling the COVID-19 virus, but the concept of trust seems less useful for explaining differences within the region.

## 5. Conclusions

In the battle against COVID-19, Denmark used many more sticks (laws and executive orders) than Sweden, where sermons (general guidelines and recommendations) were more common. This reflects a difference in the political governance and administrative structure of the two countries. Sweden had noticeably higher rates of COVID-19-related deaths and hospitalizations per 100,000 population than Denmark, yet it is difficult to determine the impact or relative effectiveness of sermons and sticks, particularly with regard to broader and longer-term health, economic and societal effects. The main policy adopters in Denmark were the government and the Danish Parliament, whereas the Public Health Agency issued most measures in Sweden. Denmark adopted strict policy measures, including lockdowns, at an early stage. Policy implementers in both countries were responsible for implementing policy measures, for monitoring and controlling compliance with the measures and for providing support to facilitate compliance with the measures. A difference was that many more of the implementers in Denmark maintained more of a monitoring and controlling function. The policy targets were broadly similar in the two countries, encompassing both the general population and specific sub-groups of the population. Similarly, there were few differences between the two countries with regard to social distancing categories included in the policy measures, although the policy measures in Denmark consisted of restrictions involving various forms of closure of parts of society to a greater extent.

## Figures and Tables

**Figure 1 ijerph-18-10990-f001:**
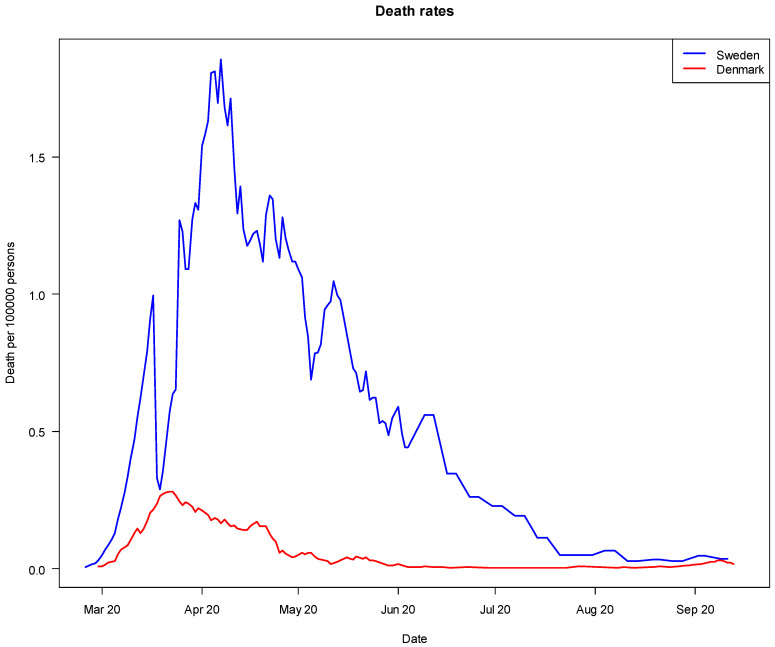
Number of COVID-19 deaths per 100,000 population in Denmark and Sweden, March 2020 to 1 October 2020 (Source: European Centre for Disease Prevention and Control 2021).

**Figure 2 ijerph-18-10990-f002:**
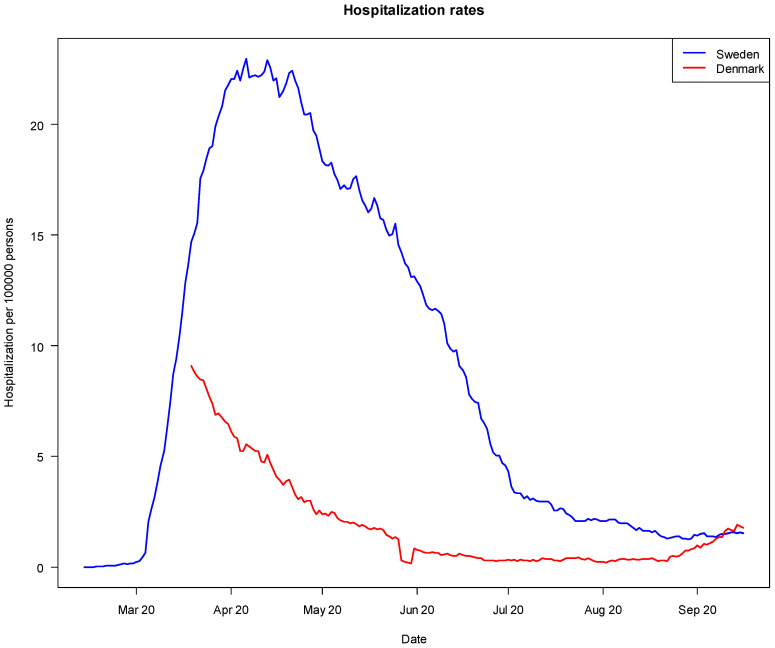
Number of COVID-19 hospitalizations per 100,000 population in Denmark and Sweden, February 2020 to 1 October 2020 (Source: European Centre for Disease Prevention and Control 2021). Notes: Data for per-day measures of hospitalization, used in Figure 2, were only available for Denmark from 2 April 2020. Data prior to this were available per-week, showing an increased hospitalization in Denmark similar to that of Sweden, followed by a decline from around April 2020. We therefore expect a similar increase in hospitalizations in Denmark as in Sweden until 2 April 2020, even though we did not have access to the per-day data.

**Table 1 ijerph-18-10990-t001:** Information about documents retrieved for the study.

	Organization	Website	Document Type
Denmark	The Danish Parliament	www.folketinget.dk (accessed on 17 October 2021)	Laws, news/press releases
	The Danish Government	www.regeringen.dk (including websites of relevant ministries) (accessed on 17 October 2021)	News/press releases, travel advice
	Retsinformation—The official legal information system of Denmark	https://www.retsinformation.dk/ (accessed on 17 October 2021)	Laws, executive orders
	The Danish Health Authority	www.sst.dk (accessed on 17 October 2021)	Recommendations, guidelines, instructions, news/press releases
	The State Serum Institute	www.ssi.dk (accessed on 17 October 2021)	News/press releases
Sweden	The Swedish Government	https://www.regeringen.se/ (including websites of relevant ministries) (accessed on 17 October 2021)	News/press releases, travel advice
	The Swedish Code of Statutes	https://svenskforfattningssamling.se (accessed on 17 October 2021)	Laws, executive orders
	The Public Health Agency in Sweden	https://www.folkhalsomyndigheten.se (accessed on 17 October 2021)	Recommendations, general guidelines, news/press releases

**Table 2 ijerph-18-10990-t002:** Aspects of the policy measure documents examined.

	Aspect	Description
1	General types of policy measures	What types of general policy measures were adopted to achieve compliance with social distancing? The measures were classified into sticks and sermons policy measures, in accordance with Vedung [17,18]
2	Categories of social distancing policy measures	What specific types of social distancing policy measures were adopted? The different types of policy measures were classified based on a taxonomy described by Rashid et al. [3]: school closure, workplace closure, home working, self-isolation of cases, quarantine of contacts, mobility restrictions, cancellation of mass events. However, further types were also explored inductively
3	Policy adopters	Who was the formal policy adopter behind the policy measure?
4	Time of adoption	When was the policy measure adopted (1 March 2020 to 1 October 2020)?
5	Policy implementers	Which organizations were responsible for implementation of the adopted policy measure?
6	Policy target group	Who are the individuals the policy measure ultimately seeks to influence?

## Data Availability

The data presented in this study are available on request from the corresponding author.

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
