# Peer review of "Social Distancing Policies in the Coronavirus Battle: A Comparison of Denmark and Sweden"

_ijerph, 2021, doi:10.3390/ijerph182010990_

Round 1

Reviewer 1 Report

The study is undoubtedly interesting, but it should be supplemented with factual data on the course of COVID-19. The conclusions should be built on comparative factual material (e.g., in terms of the number of people who have been infected, who have plagued, who have died). Then there will be an evidence base: which political system of distancing is better and more effective in neutralizing COVID-19 in Denmark or Sweden. Without this, the peer-reviewed material is of a general descriptive nature, which is available on the Internet.

Author Response

We thank the reviewer for his/her positive feedback. In the Discussion (page 12-13), we have added two figures showing the development of COVID-19 (restricted to our study period) regarding number of deaths and hospitalization per 100,000 population in Denmark and Sweden. We believe these data are the most reliable and comparable (e.g., the number of COVID-19 cases per capita would be associated with the amount of testing done). Based on these results, we have developed the discussion somewhat (in the Abstract on page 1, Discussion on page 11-14 and Conclusions on page 15) to address what impact sticks vs. sermons (Denmark and Sweden’s different strategies) might have had on the development of COVID-19 in the countries.

Reviewer 2 Report

The article can be of interest in the current context against coronavirus, it deals with the procedure of preventing the spread of the COVID 19 virus, from the perspective of public health policies in two European states.

The article lacks the methodological aspects, is not well underbuilt related to data basis (number of observations, length in time of the study, number of iterations based on observations, etc.). The results obtained by comparing the two profiles of public health policies does not reflect an impact on the population, such as significant differences and desirable means of fighting the illness with high success rates, based on case studies.

Author Response

We appreciate that the reviewer believes the study can be of interest. In the Method section (page 2-4), we have clarified methodological aspects of the study. We have added information and a table about the types of documents that were collected for the study, and from which organization and website they were retrieved. We have also clarified the data collection and analysis process, and we have specified what time period we study.

In the Discussion (page 12-13), we have added two figures showing the development of COVID-19 (restricted to our study period) regarding number of deaths and hospitalization per 100,000 population in Denmark and Sweden. We believe these data are the most reliable and comparable (e.g., the number of COVID-19 cases per capita would be associated with the amount of testing done). Based on these results, we have developed the discussion somewhat (in the Abstract on page 1, Discussion on page 11-14 and Conclusions on page 15) to address what impact sticks vs. sermons (Denmark and Sweden’s different strategies) might have had on the development of COVID-19 in the countries.

Reviewer 3 Report

The authors have done an excellent job of presenting their findings on differences in roles played by different agencies in implementing public health policies nationally. However, the readers might also be interested in knowing how the implementation of different policies by different governing agencies affected the outcomes of the pandemic in the respective countries. A few sentences on the outcome in the discussion might help understand the effectiveness of 2 different approaches. 

Overall, it is a very well-written article except for a couple of minor typographical errors. 

Author Response

We are grateful for the kind words about the study. In the Discussion (page 12-13), we have added two figures showing the development of COVID-19 (restricted to our study period) regarding number of deaths and hospitalization per 100,000 population in Denmark and Sweden. We believe these data are the most reliable and comparable (e.g., the number of COVID-19 cases per capita would be associated with the amount of testing done). Based on these results, we have developed the discussion somewhat (in the Abstract on page 1, Discussion on page 11-14 and Conclusions on page 15) to address what impact sticks vs. sermons (Denmark and Sweden’s different strategies) might have had on the development of COVID-19 in the countries.

Round 2

Reviewer 2 Report

I appreciate the authors' effort to improve the article and propose its publication.